# Cue- versus reward-encoding basolateral amygdala projections to nucleus accumbens

Yi He[1], Yanhua H Huang[2], Oliver M Schlüter[1], Yan Dong[1,2]*

[1]Departments of Neuroscience, University of Pittsburgh, Pittsburgh, United States; [2]Departments of Psychiatry, University of Pittsburgh, Pittsburgh, United States

**Abstract** In substance use disorders, drug use as unconditioned stimulus (US) reinforces drug taking. Meanwhile, drug-associated cues (conditioned stimulus [CS]) also gain incentive salience to promote drug seeking. The basolateral amygdala (BLA) is implicated in both US- and CS-mediated responses. Here, we show that two genetically distinct BLA neuronal types, expressing Rspo2 versus Ppp1r1b, respectively, project to the nucleus accumbens (NAc) and form monosynaptic connections with both dopamine D1 and D2 receptor-expressing neurons. While intra-NAc stimulation of Rspo2 or Ppp1r1b presynaptic terminals establishes intracranial self-stimulation (ICSS), only Ppp1r1b-stimulated mice exhibit cue-induced ICSS seeking. Furthermore, increasing versus decreasing the Ppp1r1b-to-NAc, but not Rspo2-to-NAc, subprojection increases versus decreases cue-induced cocaine seeking after cocaine withdrawal. Thus, while both BLA-to-NAc subprojections contribute to US-mediated responses, the Ppp1r1b subprojection selectively encodes CS-mediated reward and drug reinforcement. Such differential circuit representations may provide insights into precise understanding and manipulation of drug- versus cue-induced drug seeking and relapse.

*For correspondence: yandong@pitt.edu

Competing interest: The authors declare that no competing interests exist.

## Editor's evaluation

He et al.,2023 employ conditional genetics and in-vivo manipulations of neural activity to present valuable findings illustrating distinct functions of monosynaptic glutamatergic projections from R-spondin2-positive neurons and protein phosphatase 1 regulatory subunit 1B-positive neurons in the basolateral amygdala to the nucleus accumbens in cue-dependent operant conditioning for intracranial self-stimulation and cocaine rewards. While the evidence presented is solid and supported by well-designed experiments, the predictive capacity of the current model could be further enhanced by future experiments that delve into behavioral outcomes during operant conditioning when both types of projections are activated.

## Introduction

During the development of substance use disorder, the drug as an unconditioned stimulus (US) with its primary rewarding effects drives drug taking and seeking (*Venniro et al., 2020*; *Nader, 2016*). Meanwhile, drug use is often associated with conditioned stimuli (CS), such as environmental cues, which also acquire incentive salience to promote drug seeking and relapse (*Venniro et al., 2020*; *Nader, 2016*; *Volkow et al., 2017*). For anti-relapse treatment, an effective strategy would be to coordinately intervene the US- and CS-reinforced drug responses. However, the cellular and circuit underpinnings that differentiate the US versus CS processing are not well understood, preventing the precise and targeted manipulations.

The nucleus accumbens (NAc) is a key brain region where medium spiny neurons (MSNs) process both US and CS inputs for behavioral outputs (*Meredith et al., 2008*). Using region-specific lesion, pharmacological inhibition, and optogenetic manipulation, extensive previous studies demonstrate that the basolateral amygdala (BLA) provides both US- and CS-associated input to the NAc during drug taking and seeking (*Cardinal et al., 2002*; *Wright and Dong, 2020*; *Zinsmaier et al., 2022*). BLA pyramidal neurons heterogeneously respond to US, CS, as well as stimuli with appetitive or aversive valence (*Janak and Tye, 2015*; *O'Neill et al., 2018*; *Gründemann et al., 2019*). These BLA neurons are genetically separable into two populations, R-spondin2-positive (Rspo2) magnocellular neurons and protein phosphatase 1 regulatory subunit 1B-positive (Ppp1r1b, also known as DARPP-32) parvocellular neurons (*Kim et al., 2016*; *Kim et al., 2017*). Here, we explored whether Rspo2 and Ppp1r1b neurons project to the NAc to form two distinct BLA-to-NAc subprojections that differentially contribute to US- versus CS-reinforced drug responses.

Our results show that optogenetic stimulation of either Rspo2 or Ppp1r1b presynaptic terminals in the NAc established intracranial self-stimulation (ICSS), indicating that both subprojections contribute to US-mediated responses. However, after establishment of the subprojection-specific ICSS, only Ppp1r1b mice exhibited cue-induced ICSS seeking, suggesting that the Ppp1r1b-to-NAc subprojection selectively encodes CS-mediated reinforcement. Furthermore, chemogenetically increasing versus decreasing the transmission efficacy of Ppp1r1b-to-NAc, but not Rspo2-to-NAc, subprojection increased versus decreased cue-induced cocaine seeking after withdrawal from cocaine self-administration (SA). These results characterize the differential behavioral roles of two major BLA-to-NAc subprojections and provide a circuit mechanism through which US- versus CS-reinforced reward and drug seeking can be selectively manipulated.

## Results

### BLA Rspo2 and Ppp1r1b subprojections to the NAc

To selectively examine the Rspo2-to-NAc subprojection, we crossed the Rspo2-Cre and D1-tdTomato mice. In Rspo2-Cre × D1-tdTomato mice, we selectively expressed channelrhodopsin-2 (ChR2)-YFP in BLA Rspo2 neurons via an AAV-DIO vector and differentiated dopamine receptor D1 versus D2 NAc MSNs based on the presence versus absence of tdTomato signals (*Figure 1A–C*). Eight weeks after intra-BLA viral injection, we observed YFP signals in neuronal somas in the BLA as well as in axon fibers in the NAc core and shell (*Figure 1D–I*). We prepared brain slices containing the NAc and performed whole-cell voltage-clamp recordings of D1 and D2 MSNs in response to optogenetic stimulation of ChR2-expressing Rspo2 axon fibers (laser power, 2 mW; laser duration, 1 ms; interpulse interval, 7 s; *Figure 1J*). In both D1 and D2 MSNs, such stimulations evoked fast-activating and fast-decaying synaptic responses that were inhibited by the antagonists of glutamatergic receptors (NBQX 10 μM, D-AP5 50 μM), indicating that these synaptic responses were excitatory postsynaptic currents (EPSCs) mediated by Rspo2-to-NAc monosynaptic transmission (*Figure 1K, L*).

Ppp1r1b BLA neurons can be genetically targeted through cocaine- and amphetamine-regulated transcript protein (Cartpt)-Cre mice (*Kim et al., 2016*). We crossed Cartpt-Cre and D1-tdTomato mice to generate the mice in which we selectively expressed Ch2R in Ppp1r1b BLA neurons (*Figure 1M–O*). After 4–8 weeks, the BLA was enriched in ChR2-YFP-expressing neuronal somas, and YFP-expressing axon fibers were observed extensively in the NAc (*Figure 1P–U*). Optogenetic stimulation of these Ppp1r1b fibers evoked EPSCs in both D1 and D2 NAc MSNs (*Figure 1V–X*). Thus, both Rspo2 and Ppp1r1b BLA neurons project to the NAc and form monosynaptic connections with both D1 and D2 MSNs.

To examine the synaptic properties of Rspo2- and Ppp1r1b-to-NAc synapses, we optogenetically stimulated presynaptic terminals using a paired-pulse protocol, with interpulse intervals ranging from 25 to 200 ms (*Figure 1Y*). The paired-pulse ratio (PPR), which is inversely correlated with the presynaptic release probability (Pr), was generally lower at Ppp1r1b synapses than Rspo2 synapses, suggesting a higher basal Pr at Ppp1r1b synapses (*Figure 1Z*). Furthermore, for either Rspo2- or Ppp1r1b-to-NAc transmissions, EPSCs recorded in D1 and D2 MSNs exhibited similar PPRs, suggesting that each subprojection synapses on D1 and D2 MSNs with a similar Pr (*Figure 1Z*).

For each of the subprojections, we recorded reliable EPSCs in MSNs from both the NAc shell and core, indicating that both Rspo2 and Ppp1r1b subprojections form functional glutamatergic synapses

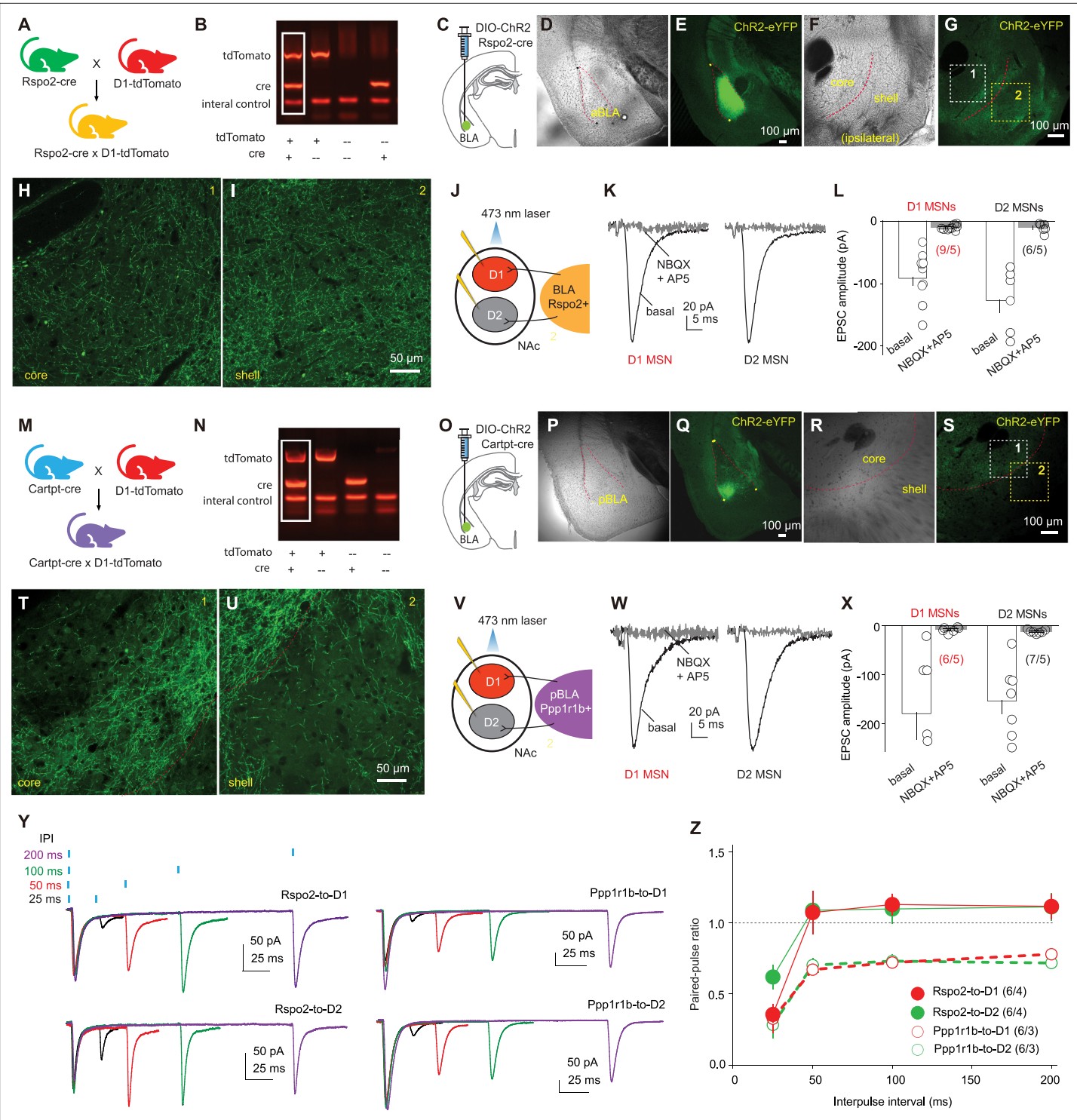

**Figure 1.** R-spondin2-positive (Rspo2)- and protein phosphatase 1 regulatory subunit 1B-positive (Ppp1r1b)-to-nucleus accumbens (NAc) projections. (**A**) Schematic for breeding Rspo2-Cre × D1-tdTomato mice using heterozygous Rspo2-Cre and D1-tdTomato mice. (**B**) Example agarose gel image showing four genotypes of bred mice. (**C**) Diagram showing injection of AAV-DIO-channelrhodopsin-2 (ChR2) into the basolateral amygdala (BLA) of Rspo2-Cre mice. (**D, E**) Example bright-field (**D**) and fluorescence (**E**) images showing ChR2-eYFP expression in the BLA. (**F, G**) Example bright-field (**F**) and fluorescence (**G**) images showing ChR2-eYFP expression in Rspo2 presynaptic terminals within the NAc. (**H, I**) Enlarged portions of (**G**) showing Ch2R-expressing fibers in the NAc core (**H**) and shell (**I**). (**J**) Diagram showing patch-clamp recordings of NAc D1 and D2 medium spiny neurons (MSNs) in response to optogenetic stimulation of Rspo2-to-NAc projection in brain slice. (**K, L**) Examples (**K**) and summary (**L**) showing that excitatory postsynaptic currents (EPSCs) evoked in D1 and D2 MSNs by optogenetic stimulation of ChR2-expressing Rspo2 presynaptic fibers were inhibited by

*Figure 1 continued*

NBQX (10 μM) and D-AP5 (50 μM) (D1, p<0.01 basal vs. NBQX+AP5; D2, p<0.01 basal vs. NBQX+AP5, two-tailed t-test). (**M**) Schematic for breeding Cartpr-Cre × D1-tdTomato mice. (**N**) Example gel image showing the genotyping of bred mice. (**O**) Diagram showing injection of AAV-DIO-ChR2 into the BLA of cocaine- and amphetamine-regulated transcript protein (Cartpt)-Cre mice. (**P, Q**) Example bright-field (**P**) and fluorescence (**Q**) images showing expression of ChR2-eYFP in the BLA. (**R, S**) Example bright-field (**R**) and fluorescence (**S**) images showing the ChR2-eYFP-expressing axon terminals in the NAc. (**T, U**) Enlarged portions of (**S**) showing Ch2R-expressing fibers in the NAc core (**T**) and shell (**U**). (**V**) Diagram showing recordings of NAc D1 and D2 MSNs in response to optogenetic stimulation of Ppp1r1b-to-NAc projection. (**W, X**) Examples (**W**) and summary (**X**) showing that EPSCs elicited in D1 and D2 MSNs by optogenetic stimulation of ChR2-expressing Ppp1r1b presynaptic fibers were inhibited by NBQX (10 μM) and D-AP5 (50 μM) (D1, p=0.01 basal vs. NBQX+AP5; D2, p<0.01 basal vs. NBQX+AP5, two-tailed t-test). (**Y**) Example EPSCs from Rspo2- or Ppp1r1b-to-NAc synapses in D1 and D2 MSNs evoked by paired-pulse stimulations with different interpulse intervals (25, 50, 100, and 200 ms). (**Z**) Summary showing lack of difference in the paired-pulse ratio between D1- and D2-MSNs within projections (Rspo2 D1 vs. Rspo2 D2, $F_{1, 10} = 0.3$, p=0.61; Ppp1r1b D1 vs. Ppp1r1b D2, $F_{1, 10} = 0.0$, p=0.83; two-way mixed ANOVA). n/m, number of recorded cells/number of mice.

The online version of this article includes the following source data for figure 1:

**Source data 1.** Source data containing numerical raw data of all panels.

on both NAc shell and core MSNs. Taken together, the Rspo2 and Ppp1r1b subprojections formed monosynaptic synapses with NAc MSNs, with presynaptic properties exhibiting between-projection difference.

## Subprojection-specific optogenetic ICSS

ICSS has been extensively used in animal models to examine the reinforcing properties of specific brain regions, neuronal types, and neural circuits. For subprojection-specific ICSS, we prepared the mice in which BLA Rspo2 or Ppp1r1b neurons expressed ChR2 such that their ChR2-expressing axon terminals in the NAc could be self-stimulated bilaterally by laser through preinstalled optical fibers (*Figure 2A*). In the operant chamber, pressing one (active), but not the other (inactive), lever resulted in a 10-pulse train of laser stimulation (pulse duration, 1 ms), and the light cue was selectively conditioned to the active lever pressing (*Figure 2A*). Thus, if the stimulation is reinforcing, ICSS will be established.

We tested the mice over several daily sessions (1 hr/session/day). In control mice in which Rspo2 or Ppp1r1b neurons expressed YFP alone without ChR2, intra-NAc laser stimulation at either 20 or 40 Hz did not induce ICSS (*Figure 2BC*). In Rspo2-ChR2 or Ppp1r1b-ChR2 mice, stimulations at 40 Hz, but not 20 Hz or lower (data not shown), established ICSS (*Figure 2DE*). Thus, strong, but not moderate, activation of the axonal fibers of Rspo2- or Ppp1r1b-to-NAc subprojection reinforces operant responding.

After the ICSS was established, we tested the mice in an extinction procedure, in which pressing the active lever resulted in presentation of light cues but no stimulation. Over 3 days of such testing, Rspo2-ChR2 mice decreased their operant response to the level as in Rspo2-YFP control mice (*Figure 2F*). In contrast, Ppp1r1b-ChR2 mice maintained high levels of responses to the active lever throughout the 3-day testing (*Figure 2G*). Thus, in addition to inducing ICSS, activation of Ppp1r1b-to-NAc subprojection may also assign incentive salience to conditioned cues, providing a circuit-based encoding for CS-mediated operant responses in the absence of US.

To determine whether this CS-mediated reinforcement is long-lasting, we retrained both Rspo2- and Ppp1r1b-ChR2 mice with ICSS for 4 additional sessions and then kept them in home cages for 21 days. After this abstinence, we placed the mice back into operant chambers for cue-induced seeking tests. Ppp1r1b-ChR2, but not Rspo2-ChR2, mice exhibited persistent cue-induced lever pressing, indicating the durable feature of the CS-mediated reinforcement established through the Ppp1r1b-to-NAc subprojection (*Figure 2HI*).

Despite ICSS-related caveats (see Discussion), the above results indicate that stimulation of either the Rspo2- or Ppp1r1b-to-NAc subprojection may serve as a US for establishing ICSS, while the Ppp1r1b subprojection functions as a key component within the BLA-to-NAc input that encodes CS-mediated reinforcement. These differences between the two subprojections predict that they play differential roles in cue-conditioned drug seeking, which was tested below.

## Acquisition and performance of cocaine SA

The BLA-to-NAc projection is one of the primary circuit underpinnings for cue-conditioned cocaine seeking (*Wright and Dong, 2020*; *Zinsmaier et al., 2022*). We first examined whether the two BLA

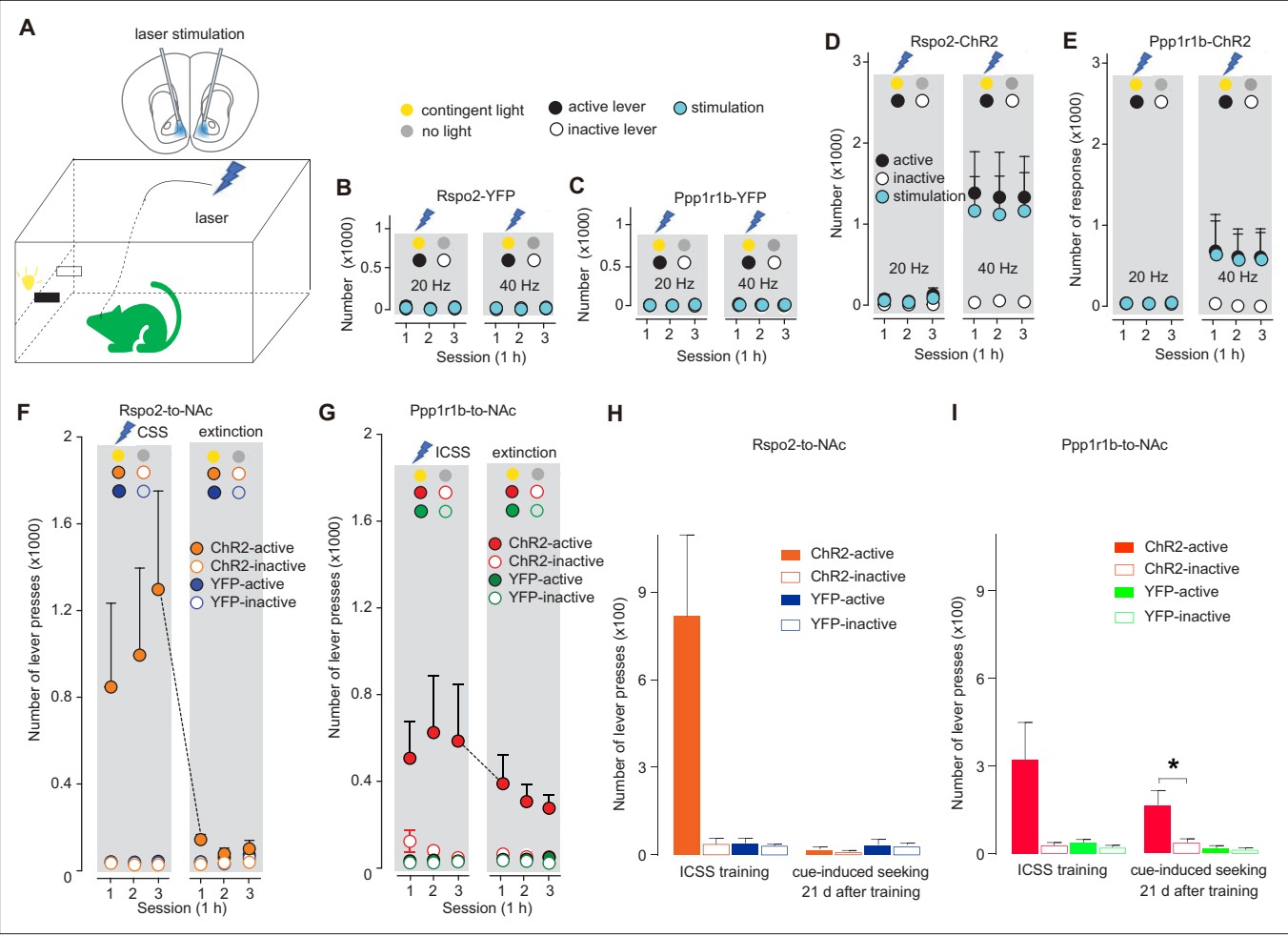

**Figure 2.** Optogenetic intracranial self-stimulation (ICSS) of R-spondin2-positive (Rspo2)- and protein phosphatase 1 regulatory subunit 1B-positive (Ppp1r1b)-to-nucleus accumbens (NAc) presynaptic terminals. (**A**) Diagrams showing intra-NAc optogenetic stimulation of Rspo2- or Ppp1rb-to-NAc presynaptic terminals during ICSS (upper) and the optogenetic ICSS setup (lower). (**B, C**) Summaries showing that intra-NAc stimulation at 20 or 40 Hz did not establish ICSS in control mice, in which basolateral amygdala (BLA) Rspo2-to-NAc (active vs inactive, $F_{1,6}$ = 0.1, p=0.76, two-way mixed ANOVA, **B**) or Ppp1r1b-to-NAc (active vs inactive, $F_{1,6}$ = 0.0, p=0.88, two-way mixed ANOVA, **C**) presynaptic terminals expressed YFP. (**D, E**) Summaries showing that intra-NAc optogenetic stimulation of channelrhodopsin-2 (ChR2)-expressing Rspo2- (20 Hz, active vs inactive, $F_{1,6}$ = 2.8, p=0.14, two-way mixed ANOVA; 40 Hz, active vs inactive, $F_{1,6}$ = 6.3, p=0.04, two-way mixed ANOVA, **D**) or Ppp1r1b-to-NAc (20 Hz, active vs inactive, $F_{1,6}$ = 0.9, p=0.37, two-way mixed ANOVA; 40 Hz, active vs inactive, $F_{1,6}$ = 2.6, p=0.16, two-way mixed ANOVA, **E**) presynaptic terminals at 40 Hz, but not 20 Hz, established ICSS. (**F, G**) Summaries showing that, compared with Rspo2-ChR2 mice (YFP vs ChR2 main effect, $F_{1,12}$ = 2.1, p=0.17, three-way ANOVA, **F**) Ppp1r1b-ChR2 mice (YFP vs ChR2 main effect, $F_{1,12}$ = 8.9, p=0.01, three-way ANOVA, **G**) exhibited persistent cue-induced operant ,responses during the extinction test (lever press resulting in cue presentation without optogenetic stimulation) after cue-conditioned ICSS was established. (**H, I**) Summaries showing that after 21-day abstinence following the ICSS establishment, cue-induced responding to the active lever was minimal in Rspo2-ChR2 mice (ChR2-active vs ChR2-inactive, t=0.8, p=0.48, n=4, two-tailed t-test, **H**) but robust in Ppp1r1b-ChR2 mice (ChR2-active vs ChR2-inactive, t=2.9, p=0.03, n=4, two-tailed t-test, **I**).

The online version of this article includes the following source data for figure 2:

**Source data 1.** Source data containing numerical raw data of all panels.

subprojections contribute to the acquisition of cocaine SA, a procedure through which contextual and discrete cues are conditioned to cocaine experience. Our experimental strategy was to examine whether experimentally enhancing or decreasing the transmission of a subprojection influences cocaine taking during cocaine SA.

To enhance the transmission efficacy, we selectively expressed a stabilized step function opsin (SSFO) in Rspo2 or Ppp1r1b BLA neurons and installed optical fibers in the NAc (*Figure 3AB*). Activation of SSFO upon 473 nm laser stimulation generates a low but persistent (deactivation tau at ~29 min) conductance to cations (*Yizhar et al., 2011*). With these features, activation of SSFO does

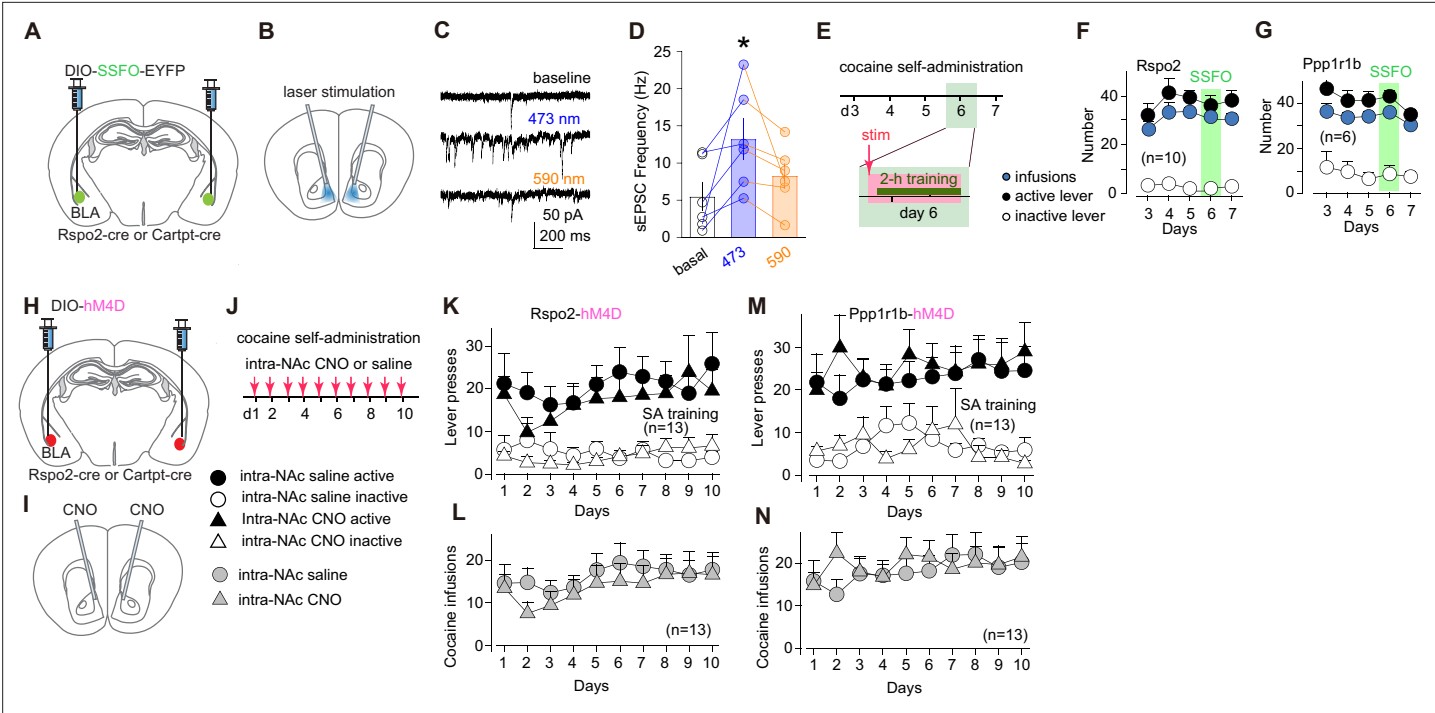

**Figure 3.** Roles of R-spondin2-positive (Rspo2)- and protein phosphatase 1 regulatory subunit 1B-positive (Ppp1r1b)-to-nucleus accumbens (NAc) transmissions in acquisition of cocaine self-administration. (**A, B**) Diagrams showing viral expression of stabilized step function opsin (SSFO) in basolateral amygdala (BLA) Rspo2 or Ppp1r1b neurons (**A**) and intra-NAc optical stimulation (**B**). (**C**) Example spontaneous excitatory postsynaptic currents (sEPSCs) in NAc medium spiny neurons (MSNs) during baseline recording, after 473 nm laser stimulation, and after 590 nm laser stimulation of SSFO-expressing presynaptic terminals. (**D**) Summaries showing that stimulation of SSFO-expressing presynaptic terminals by 473 nm laser induced an increase in sEPSC frequency, and this increase was reversed by 590 nm laser ($F_{2,10}$ = 6.0, p=0.02, one-way ANOVA repeated measure). (**E**) Timeline for SSFO-mediated manipulation of Rspo2- or Ppp1r1b-to-NAc transmission during cocaine self-administration. (**F, G**) Summaries showing that numbers of active lever presses and optical stimulations during cocaine self-administration were not altered upon SSFO-mediated upregulation of Rspo2-to-NAc (active lever press, $F_{4,36}$ = 1.2, p=0.32; inactive lever press, $F_{4,36}$ = 1.5, p=0.22; infusion, $F_{4,36}$ = 1.5, p=0.21; one-way ANOVA repeated measure, **F**) or Ppp1r1b-to-NAc (active lever press, $F_{4,20}$ = 1.5, p=0.25; inactive lever press, $F_{4,20}$ = 0.4, p=0.87; infusion: $F_{4,20}$ = 1.8, p=0.16, Tukey posttest, **G**), compared to data 1 day before and 1 day after the day with SSFO stimulation. (**H, I**) Diagrams showing viral expression of hM4D DREADDs in BLA Rspo2 or Ppp1r1b neurons (**H**) and intra-NAc infusion of clozapine N-oxide (CNO) (**I**). (**J**) Experimental schematic of intra-NAc infusion of CNO or saline control during daily cocaine self-administration. Inset: Symbols of mice with different manipulations in **K–N**. (**K, L**) Summaries showing that chemogenetic inhibition of Rspo2-to-NAc transmission did not alter numbers of active lever presses (saline vs CNO, $F_{1,24}$ = 0.4, p=0.53, two-way mixed ANOVA, **K**) or numbers of cocaine infusions (saline vs CNO, $F_{1,24}$ = 0.4, p=0.52, two-way mixed ANOVA, **L**). (**M, N**) Summaries showing that chemogenetic inhibition of Ppp1r1b-to-NAc transmission did not alter numbers of active lever presses (saline vs CNO, $F_{1,24}$ = 0.3, p=0.61, two-way mixed ANOVA, **M**) or numbers of cocaine infusions (saline vs CNO, $F_{1,24}$ = 0.1, p=0.71, two-way mixed ANOVA, **N**).

The online version of this article includes the following source data for figure 3:

**Source data 1.** Source data containing numerical raw data of all panels.

not evoke action potentials but instead moderately depolarizes the membrane potential to facilitate presynaptic release (*Yizhar et al., 2011*). After viral-mediated expression of SSFO in the prefrontal cortex or BLA, our previous studies verify that laser stimulation (473 nm, 50 ms × 10 pulses at 10 Hz) turns on SSFO, which persistently (>40 min) and selectively facilitates presynaptic release from these projections to NAc MSNs without generating back-propagating action potentials (*Wang et al., 2020*; *Liu et al., 2016*). To further verify this SSFO approach, we prepared NAc slices containing SSFO-expressing BLA presynaptic terminals and recorded spontaneous EPSCs (sEPSCs) in MSNs. A 473 nm laser train (10 pulses at 10 Hz) of stimulation increased the frequency ($F_{2,10}$ = 6.0, p=0.02) without affecting the amplitude ($F_{2,10}$ = 1.3, p=0.33) of sEPSCs, and this effect of SSFO was terminated upon application of a 590 nm laser train (10 pulses at 10 Hz), indicating an effective and reversible presynaptic upregulation by SSFO (*Figure 3CD*).

After 8 weeks of viral expression, we trained the mice with a 7-day cocaine SA procedure (0.75 mg/kg/infusion, 2 hr/day, fixed ratio 1 schedule). Mice established cocaine SA over the first 2 days and

their daily responses to active lever stabilized afterward. On day 6, we applied intra-NAc laser stimulation bilaterally to induce enhanced presynaptic release of the Rspo2- or Ppp1r1b-to-NAc transmission and then placed the mice to the operant chamber for cocaine SA training (*Figure 3E*). With this manipulation, neither the response to active lever nor cocaine intake was altered in Rspo2 or Ppp1r1b mice on day 6, compared to the day before or after (*Figure 3F, G*). Thus, upregulation of Rspo2- or Ppp1r1b-to-NAc transmission does not affect the performance of cocaine SA.

To decrease the transmission efficacy, we selectively expressed inhibitory DREADDs (hM4D) in Rspo2 or Ppp1r1b BLA neurons and infused 500 nL of clozapine *N*-oxide (CNO) (3 µM) into the NAc ~15 min before the SA training on each training day (*Figure 3H–J*). As such, the activity of Rspo2 or Ppp1r1b presynaptic terminals within the NAc was locally decreased. Over 10 days of cocaine SA, both Rspo2-CNO and Ppp1r1b-CNO mice exhibited similar levels of active lever responding and cocaine intake compared to control mice, which expressed DREADDs but with daily intra-NAc infusion of 500 nL of saline (*Figure 3K–N*). Thus, downregulation of Rspo2- or Ppp1r1b-to-NAc transmission does not affect the acquisition or performance of cocaine SA.

Taken together, neither the Rspo2- nor Ppp1r1b-to-NAc subprojection seems to be essential for acquiring and maintaining the operant response for cocaine, leading our subsequent experiments to explore the role of these subprojections in cue-conditioned responses after cocaine SA was established.

## Subprojection-specific regulation of cue-induced cocaine seeking

Through cocaine SA, conditioned cues acquire reinforcing properties that drive cocaine seeking. We next examined whether the two BLA subprojections regulated cue-induced cocaine seeking after cocaine SA was established. We prepared the mice such that excitatory (hM3D) or inhibitory (hM4D) DREADDs was selectively expressed in Rspo2 versus Ppp1r1b neurons in the BLA bilaterally for subsequent chemogenetic manipulations of these two subprojections in vivo (*Figure 4A–L*).

Without chemogenetic manipulations, mice with intra-BLA expression of hM4D were trained with 10-day cocaine SA to establish stable operant responding to the active lever for cocaine intake (*Figure 4M, N, P, Q*). On withdrawal days 7, 21, and 45, mice received intra-NAc injections of CNO or saline control and were placed into operant chambers immediately. Rspo2 mice with hM4D-CNO exhibited similar levels of cue-induced cocaine seeking compared to the hM4D-saline control mice (*Figure 4O*). However, cue-induced cocaine seeking was decreased in Ppp1r1b mice with hM4D-CNO, compared to hM4D-saline controls (*Figure 4R*).

For hM3D-expressing mice, we also trained them with 10-day cocaine SA and tested cocaine seeking on withdrawal days 7, 21, and 45 (*Figure 4STVW*). While hM3D-CNO and hM3D-saline Rspo2 mice exhibited similar levels of cue-induced cocaine seeking (*Figure 4U*), cue-induced cocaine seeking was increased in hM3D-CNO Ppp1r1b mice (*Figure 4X*).

These results suggest the Ppp1r1b subprojection as a key component of the BLA-to-NAc circuit that regulates the behavioral expression of cue-conditioned cocaine seeking.

## Discussion

Our current study characterizes the basic circuit and behavioral properties of Rspo2- and Ppp1r1b-to-NAc subprojections in the context of cocaine seeking and relapse. The two BLA-to-NAc subprojections may be part of the circuit mechanism through which US- and CS-mediated reinforcements are differentially encoded and, thus, can be targeted for selective manipulation of drug- versus cue-induced drug relapse.

### The two BLA-to-NAc subprojections

The BLA projection to NAc is critically implicated in both US- and CS-mediated reinforcement, including drug- and cue-induced drug seeking (*Everitt and Robbins, 2005*; *Puaud et al., 2021*). In behaving animals, BLA neurons tend to be activated in a population-specific manner in response to US versus CS as well as stimuli with different valences, suggesting the functional and populational heterogeneity of BLA neurons (*O'Neill et al., 2018*; *Paton et al., 2006*; *Tye and Janak, 2007*). Based on morphological and anatomical properties, BLA pyramidal neurons can be divided into two populations, magnocellular versus parvocellular neurons that are preferentially located in the anterior versus posterior

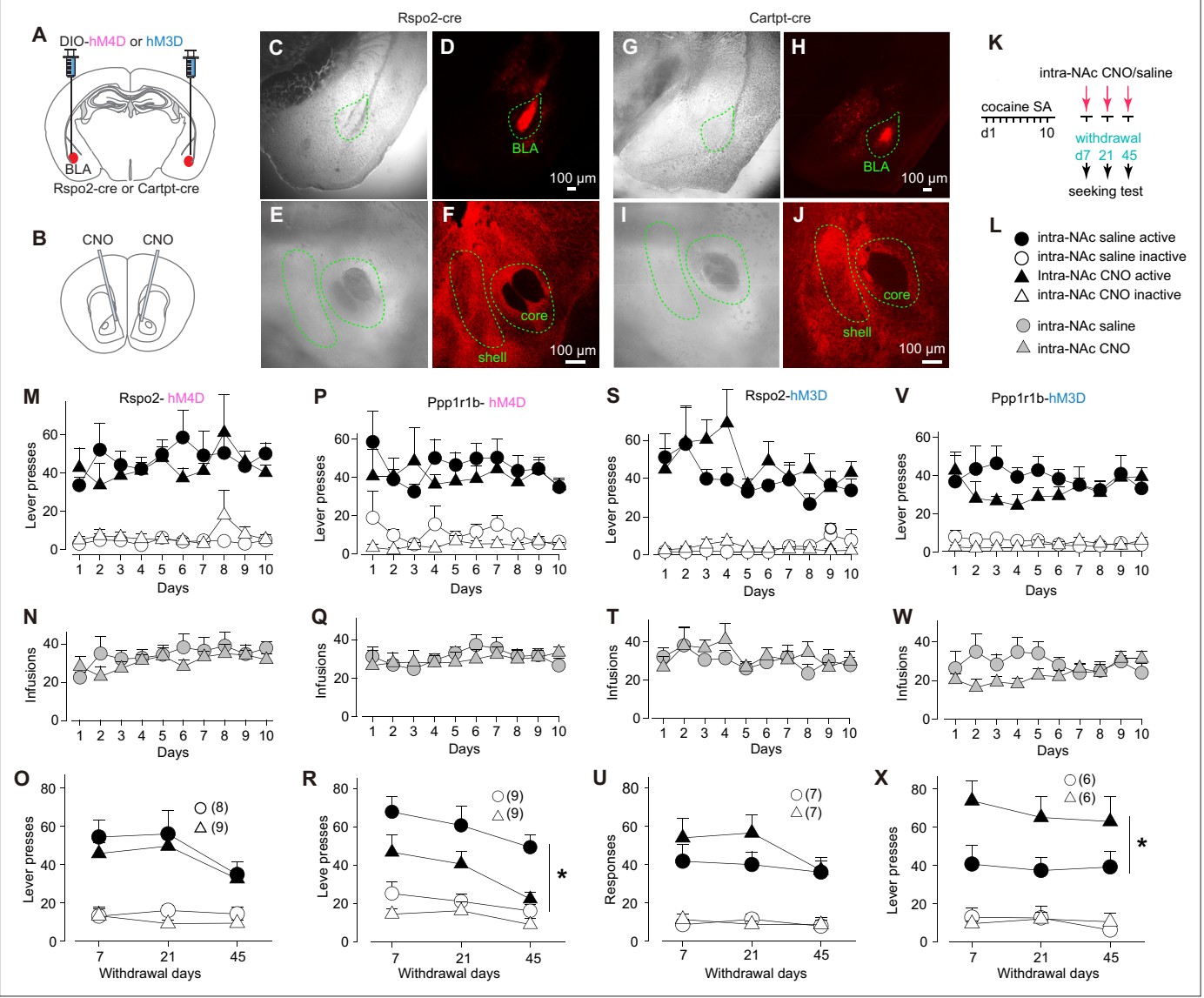

**Figure 4.** Roles of protein phosphatase 1 regulatory subunit 1B-positive (Ppp1r1b)- and R-spondin2-positive (Rspo2)-to-nucleus accumbens (NAc) transmissions in cue-induced cocaine seeking. (**A, B**) Diagram showing viral expression of DREADDs in basolateral amygdala (BLA) Rspo2 or Ppp1r1b neurons (**A**) and intra-NAc application of clozapine *N*-oxide (CNO) (**B**). ( **C–J**) Example bright-field and fluorescence images of the BLA and NAc in mice with Cre-dependent intra-BLA expression of DREADDs in Rspo2-Cre (**C–F**) and cocaine- and amphetamine-regulated transcript protein (Cartpt)-Cre (**G–J**) mice. (**K**) Schematic of cocaine self-administration and DREADDs-mediated manipulation of Rspo2- or Ppp1r1b-to-NAc subprojection during cue-induced cocaine seeking after withdrawal from cocaine self-administration. (**L**) Collection of symbols indicating mice with different manipulations in **M–X**. (**M–R**) Summaries showing that, after 10 days of cocaine self-administration, cue-induced cocaine seeking (assessed by numbers of active lever presses) was not altered in Rspo2-hM4D mice (**M–O**) but decreased in Ppp1r1b-hM4D (**P–R**) mice upon intra-NAc application of CNO (Rspo2: saline vs CNO, $F_{1,15} = 0.6$, p=0.45; Ppp1r1b: saline vs CNO, $F_{1,10} = 7.1$, p=0.02; two-way mixed ANOVA main effect). (**S–X**) Summaries showing that, after 10 days of cocaine self-administration, cue-induced cocaine seeking was not altered in Rspo2-hM3D mice (**M–O**) but increased in Ppp1r1b-hM3D (**P–R**) mice upon intra-NAc application of CNO (Rspo2: saline vs CNO, $F_{1,12} = 1.6$, p=0.23; Ppp1r1b: saline vs CNO, $F_{1,10} = 7.4$, p=0.02; two-way mixed ANOVA main effect).

The online version of this article includes the following source data for figure 4:

**Source data 1.** Source data containing numerical raw data of all panels.

BLA, respectively (*Swanson and Petrovich, 1998*; *McDonald, 1982*). Genetic profiling reveals that these two neuronal types correspond to Rspo2- and Ppp1r1b-expressing neurons, which are differentially implicated in appetitive versus aversive affective responding (*Kim et al., 2016*; *Kim et al., 2017*). Our current results show that both Rspo2 and Ppp1r1b BLA neurons project to the NAc, forming

monosynaptic connections with both D1 and D2 MSNs (*Figure 1*). In our experiments, stimulation of Rspo2- or Ppp1r1b-to-NAc presynaptic fibers evokes EPSCs in randomly sampled MSNs, suggesting co-innervation of individual MSNs by both BLA subprojections. However, the low-throughput feature of electrophysiology did not allow to generate large sample sizes and, thus, did not rule out other possible innervation patterns. Nonetheless, if co-innervation is true, each MSN may function as a basic NAc unit to integrate Rspo2 and Ppp1r1b inputs. Potentially through different downstream targets, D1 and D2 NAc MSNs play differential and, on many occasions, opposing roles in cue-conditioned responses (*Zinsmaier et al., 2022*; *Smith et al., 2013*). In conditioned responses, the driving force and inhibitory control are the two essential components that cooperate to ensure a smooth behavioral output. We speculate that the co-innervation of D1 and D2 MSNs by Rspo2 or Ppp1r1b provides a circuit basis on which the dynamic balance between the drive and inhibition can be achieved.

A prominent functional difference between the Rspo2- and Ppp1r1b-to-NAc synapses is their different PPRs (*Figure 1*). Because both subprojections are assessed by stimulation of ChR2-expressing presynaptic terminals with the same set of parameters, potential systemic errors are normalized. These results suggest that the presynaptic Pr of Ppp1r1b synapses is higher than Rspo2 synapses. In response to CS, BLA neurons fire action potentials with heterogenous patterns (*Ambroggi et al., 2008*). With these synaptic features, the Rspo2- versus Ppp1r1b-to-NAc transmissions are expected to be more effective in activating NAc MSNs upon bursting versus tonic activations, respectively.

## ICSS

Optogenetic stimulation of either Rspo2 or Ppp1r1b presynaptic terminals within the NAc establishes ICSS, revealing the reinforcing capacity of these two subprojections (*Figure 2*). However, recent results show that Rspo2 and Ppp1r1b BLA neurons may encode opposite valence such that they selectively express activity biomarkers in response to aversive versus appetitive stimuli, respectively (*Kim et al., 2016*; *Kim et al., 2017*). Despite the different experimental approaches, these seemingly discrepant results may reveal the projection-specific functions of Rspo2 neurons. Specifically, results from the retrograde tracing show that, compared with the NAc, a larger portion of Rspo2 neurons send much denser projections to the central nucleus of amygdala (CeA), a brain region associated with negative behaviors (*Kim et al., 2016*). BLA neurons that project to the CeA versus NAc are preferentially activated by aversion- versus reward-predictive cues, and activation of BLA-to-CeA versus BLA-to-NAc promotes aversive versus appetitive behaviors (*Kim et al., 2017*; *Beyeler et al., 2016*). We speculate that Rspo2 BLA neurons projecting to the CeA versus NAc are largely nonoverlapping, mediating distinct behavioral responses.

Without differentiating Rspo2- or Ppp1r1b-to-NAc subprojections, previous studies show that mice establish instrumental responding for self-stimulation of the BLA-to-NAc projection as a whole (*Stuber et al., 2011*). Taken together with our current results, activation of each of the BLA-to-NAc subprojections may function as an unconditioned reinforcer to drive the instrumental responding. However, an interpretational caveat should be considered. Specifically, low-frequency (i.e., 5 or 10 Hz over ~5 s) stimulations of BLA neurons do not establish ICSS (*Servonnet et al., 2020*). Similarly in our experiments, relatively low-frequency (i.e., 20 Hz) optogenetic stimulation of either of the subprojections does not establish ICSS; ICSS is established only when the stimulation frequency is increased to 40 Hz (*Figure 2*). Through potential backpropagation, such high-frequency stimulations of presynaptic terminals can strongly activate the soma of BLA neurons, which, in turn, activate other BLA-projected brain regions that are involved in US-mediated reinforcement. Indeed, direct or indirect BLA projections are detected to innervate neurons in the ventral tegmental area, prefrontal cortex, hippocampus, and lateral hypothalamus, all implicated in behavioral reinforcement (*Janak and Tye, 2015*). Thus, in addition to BLA-to-NAc subprojections, other divergent projections from the BLA may also be activated to substantiate the US-mediated reinforcement during ICSS training.

It has long been known that the BLA-to-NAc projection is essential for cue-conditioned incentive processing (*Robinson and Berridge, 1993*; *Chang et al., 2012*; *Keefer et al., 2021*). Our results show that stimulation of Ppp1r1b-, but not Rspo2-to-NAc, subprojection, induced a prolonged cue-conditioned responding, highlighting the Ppp1r1b subprojection in cue conditioning (*Figure 2*). These results are consistent with the topographic features of the BLA-to-NAc projection. Specifically, compared to the anterior BLA, the posterior BLA is more enriched in Ppp1r1b neurons and send projections to both the NAc shell and core (*Kim et al., 2016*; *Kita and Kitai, 1990*; *Russchen and Price,*

*1984*; *Brog et al., 1993*). Disconnecting the BLA and NAc shell impairs outcome-specific Pavlovian-to-instrumental transfer and the maintenance of cue-conditioned responding in the absence of reward (*Simmons and Neill, 2009*; *Shiflett and Balleine, 2010*; *Corbit and Balleine, 2011*). Thus, among the NAc sell and core projections, the Ppp1r1b-to-NAc shell branch may be particularly important for cue-conditioned reinforcement.

### Acquisition and expression of cue-induced cocaine seeking

Our results show that manipulating either of the BLA subprojections does not affect the acquisition of cocaine SA (*Figure 3*). This is in line with the findings that lesion of the BLA does not prevent the acquisition of instrumental responding to rewards (*Keefer et al., 2021*; *Balleine et al., 2003*; *Holland and Gallagher, 2003*; *Wassum and Izquierdo, 2015*). Indeed, the BLA is also not essential for first-order cue conditioning under many experimental conditions (*Hatfield et al., 1996*; *Blundell et al., 2001*; *Parkinson et al., 2000*). Albeit nonessential, our ICSS results suggest that activation of the Ppp1r1b subprojection is sufficient to establish cue conditioning (*Figure 2*). Thus, BLA inputs to the NAc are likely among several parallel and potentially redundant circuits that collectively contribute to cue conditioning during motivated behaviors.

After withdrawal from cocaine SA, chemogenetically enhancing versus inhibiting the Ppp1r1b-to-NAc subprojection increases versus decreases cue-induced cocaine seeking, respectively (*Figure 4*). Thus, instead of the acquisition (*Figure 3*), the Ppp1r1b subprojection preferentially regulates the execution of cue-reinforced cocaine seeking. This is consistent with the previous results that disconnection and chemo- or optogenetic inhibition of BLA-to-NAc projection compromise the performance of cue-controlled reward seeking (*Ambroggi et al., 2008*; *Stuber et al., 2011*; *Di Ciano and Everitt, 2004*), including cue-induced cocaine seeking under a second-order schedule of reinforcement (*Puaud et al., 2021*). Our current study extends these findings by pinpointing the Ppp1r1b subprojection as the key circuit component within the BLA-to-NAc projection for cue-induced cocaine responses.

Emerging evidence suggests that the BLA-to-NAc projection is one of the primary circuit targets for drug experience, with the resulting adaptive changes promoting drug seeking. For example, after withdrawal from cue-conditioned cocaine SA, the overall strength of randomly sampled BLA-to-NAc synapses is increased, and reversing this cocaine-induced strengthening decreases cue-induced cocaine seeking (*Zinsmaier et al., 2022*; *Pascoli et al., 2014*; *Lee et al., 2013*). In these studies, the behavioral reversal is observed upon manipulation of BLA inputs to the NAc shell. Among the heterogenous BLA-to-NAc projections, we speculate that the Ppp1r1b subprojection is the primary bearer of cocaine-induced adaptations that promote cue-induced cocaine seeking.

By manipulating somatic activities or projections to the CeA, previous studies demonstrate opposite roles of Rspo2 and Ppp1r1b BLA neurons in appetitive versus aversive reinforcement (*Kim et al., 2016*; *Kim et al., 2017*). Our chemogenetic manipulation above was confined to the NAc presynaptic terminals of Rspo2 BLA neurons. As such, a lack of behavioral effects in our results may suggest that the established behavioral correlates of Rspo2 BLA neurons are mediated by the projections of Rspo2 BLA to other brain regions.

Taken together, our current study pinpoints the differential roles of two major BLA-to-NAc subprojections in regulating US- versus CS-mediated reinforcement. These results may provide a circuit basis through which drug- and drug-conditioned cues invigorate drug craving and seeking.

## Materials and methods
### Subjects and reagents

Both Rspo2-Cre and Cartpt-Cre mice with the C57BL/6J background were obtained from the Tonegawa lab (*Kim et al., 2016*). The D1-tdTomato mouse line (strain number 016204) was purchased from the Jackson Lab (Bar Harbor, ME, USA) and bred in house. The two Cre lines were crossed with the D1-tdTomato line to generate Rspo2-Cre-D1-tdTomato and Ppp1r1b-Cre-D1-tdTomato mice.

Male mice were used in all experiments. Mice were grouped and allocated for planed experiments at the age of 6–8 weeks, and were singly housed on a regular 12 hr light/dark cycle (light on/off at 7:00/19:00), with food and water available ad libitum. After acclimation, surgery, and post-surgery recovery, mice at ~3-month-old were used to prepare brain slices for electrophysiology recording.

The animal use was in accordance with the NIH guideline and under protocols approved by the Institutional Animal Care and Use Committees at the University of Pittsburgh.

AAV2-EF1a-DIO-hCHR2(C128S/D156A)-EYFP (AAV-DIO-SSFO), AAV9-EF1a-DIO-hChR2(H134R)-EYFP-WPRE-pA (AAV-DIO-ChR2), and control virus AAV9-EF1a-DIO-EYFP-WPRE-pA (AAV-DIO-EYFP) were purchased from the University of North Carolina Vector Core. AAV9-hSyn-DIO-hM4D(Gi)-mCherry (AAV-DIO-hM4D), AAV9-hSyn-DIO-hM3D(Gq)-mCherry (AAV-DIO-hM3D), and control virus AAV9-hSyn-DIO-mCherry (AAV-DIO-mCherry) were purchased from Addgene (Watertown, MA, USA). All chemicals used for electrophysiological recording were purchased from Sigma-Aldrich or Thermo Fisher Scientific, except for D-AP5 and NBQX, which were obtained from Alomone Labs (Jerusalem, Israel). Cocaine-HCl was supplied by the Drug Supply Program of the National Institute on Drug Abuse. The DREADD agonist CNO dihydrochloride (water-soluble) was obtained from Hello Bio Inc (Princeton, NJ, USA).

## Genotyping
The genotype of mice was determined by using the Extract-N-Amp Tissue PCR Kit from Sigma-Aldrich (St Louis, MO, USA). The primer sequences (5′→3′) for targeted gene and reference gene were: *tdTomato* (Forward: ACAACATGGCCGTCATCA, Reverse: ACAGCTCGTCCATGCCGTA), *Cre* (Forward: AATGCTTCTGTCCGTTTGCC, Reverse: GATCCGCCGCATAACCAGT), *Alas1* (Forward: TCTTCACC ACCTCCTTGCCAC, Reverse: AGGCTTTCTCTCTTTCGCTCA). Touch-down PCR cycling was used, and temperatures were optimized for simultaneously amplifying multiple genes. The genotype of the samples was analyzed by gel electrophoresis. The expected length of PCR products of tdTomato, Cre, and Alas1 genes were 737, 232, and 115 bp, respectively.

## In vivo viral injection
Mice were anesthetized with i.p. injection of ketamine (100 mg/kg)-xylazine (10 mg/kg) mixture and were held in a stereotaxic frame (Stoelting Co., Wood Dale, IL, USA). Through a 10 µL NanoFil syringe with a 33-gauge needle controlled by UMP3 and Micro4 system (WPI, Sarasota, FL, USA), the AAV solution (500 nL per side) was infused bilaterally at a rate of 100 nL/min into the BLA of Rspo2-Cre mice (in mm: to bregma AP, −1.6; ML, ±3.3; DV, −5.0) or BLA of Cartpt-Cre mice (AP, −2.0; ML, ±3.4; DV, −5.0) (*Kim et al., 2016*; *Yu et al., 2022*). The needles were held in place for 5 min before removal to minimize the backflow of viral solutions. After surgery, mice were placed on a heating pad for recovery. Carprofen (5 mg/kg) was injected (s.c.) daily for up to 3 days after surgery. Mice were then kept in their home cages for >6 weeks for viral expression before electrophysiology recording or additional surgeries for behavioral tests.

## Slice preparation for electrophysiology
To prepare acute brain slices, mice were decapitated under isoflurane anesthesia. Sagittal slices (250 µm) containing the NAc were prepared on a VT1200S vibratome (Leica) in a 4°C cutting solution containing (in mM) 135 *N*-methyl-d-glucamine, 1 KCl, 1.2 $KH_2PO_4$, 0.5 $CaCl_2$, 1.5 $MgCl_2$, 20 choline-$HCO_3$, and 11 glucose, saturated with 95% $O_2$/5% $CO_2$, pH adjusted to 7.4 with HCl. Osmolality was adjusted to 300. Slices were incubated in the artificial cerebrospinal fluid, containing (in mM): 119 NaCl, 2.5 KCl, 2.5 $CaCl_2$, 1.3 $MgCl_2$, 1 $NaH_2PO_4$, 26.2 $NaHCO_3$, and 11 glucose, with the osmolality adjusted to 290–295, saturated with 95% $O_2$/5% $CO_2$. The brain slices were incubated at 34°C for 30 min and then allowed to recover for >30 min at 20–22°C before experimentation.

## Optogenetic electrophysiology
We recorded optogenetically evoked EPSCs in a projection- and cell type-specific manner (*Yu et al., 2022*; *Yu et al., 2019*). Voltage-clamp whole-cell recordings were performed using an Axon Multi-Clamp 700B amplifier (Molecular Devices, San Jose, CA, USA) and borosilicate glass pipettes pulled on a P-97 Puller (Sutter Instruments, Novato, CA, USA), with the resistance of 2–5 MΩ when filled with pipette a cesium-based internal solution containing (in mM): 140 $CsMeSO_3$, 5 TEA-Cl, 0.4 EGTA (Cs), 20 HEPES, 2.5 Mg-ATP, 0.25 Na-GTP, and 1 QX-314 (Br), with pH adjusted to 7.2–7.3 and osmolarity to 275 mOsm (*Wang et al., 2021*). The 473 nm blue light was delivered to ChR2-expressing presynaptic fibers by a DPSS laser source obtained from IkeCool Corporation (Anaheim, CA, USA), which was coupled to a fluorescent port of the Olympus BX51WI microscope through 62.5 µm optic fiber,

allowing the blue laser to illuminate through the objective. Clampex software 9.2 (Molecular Devices, San Jose, CA, USA) was used for triggering laser stimulation and data acquisition. Synaptic currents were recorded, filtered at 2.6–3 kHz, amplified five times, and then digitized at 20 kHz. Data were analyzed and plotted using Clampfit 9.2.

To assess whether the postsynaptic currents recorded in D1 and D2 MSNs are mediated by BLA-to-NAc glutamatergic transmission, recordings were performed at –70 mV, with stimulation of a single laser pulse (duration <1 ms) every 7 s for 30 times. Subsequently, a cocktail of NBQX (10 µM) and D-AP5 (50 µM) was bath-applied to confirm if the responses were mediated by AMPARs and NMDARs. To assess the presynaptic release probability, optogenetically evoked EPSCs were measured by paired-pulse stimulations with interpulse intervals of 25, 50, 100, or 200 ms, repeatedly delivered every 10 s. The PPR was calculated as the ratio of the amplitude of the second response over the first one.

## Stereotaxic cannula implantation

To perform in vivo laser stimulation of SSFO and optogenetic intracranial self-stimulation (oICSS), the optical fiber cannulae (0.22 NA, Ø200 µm, Thorlabs, Inc, Newton, NJ, USA) surrounded by ceramic ferrules (Ø1.25 mm, Ø200 µm Core, Thorlabs, Inc, Newton, NJ, USA) were bilaterally inserted into the NAc of Cartpt-Cre mice (AP, 1.1; ML, ±1.8; DV, −4.2; at 10° lateral angle) or Rspo2-Cre mice (AP, 1.5; ML, ±1.3; DV, −4.3; at 10° lateral angle). Dental cement and screws were used to secure the cannulae on the skull.

In experiments involving DREADDs, the mice were implanted bilaterally with the guide cannulae (C315GMN/SPC, P1 Technologies, Roanoke, VA, USA) in the NAc 1 week prior to cocaine SA. The guide cannulae were stereotaxically lowered into the NAc of Cartpt-Cre mice (AP, 1.1; ML, ±1.8; DV, −3.2; at 10° lateral angle) and Rspo2-Cre mice (AP, 1.5; ML, ±1.3; DV, −3.3; at 10° lateral angle). After being secured to the skull using dental cement and screws, the cannulae were sealed by cannula dummies (C315DCMN/SPC, P1 Technologies, Roanoke, VA, USA), projected 1 mm beyond the length of the guide cannula.

## Laser stimulation of SSFO

The projection-specific stimulation was conducted bilaterally in Rspo2-Cre or Cartpt-Cre mice with AAV-DIO-SSFO expression through optical fibers in the NAc. The 473 nm laser was applied by a laser generator (Shanghai Laser & Optics Century Co., Ltd. CA), controlled by a Master-8 pulse generator (AMPI, Jerusalem, Israel). The light power at the tip of the fiber was measured with a power meter (PM20CH, Thorlabs, Inc, Newton, NJ, USA) prior to experiments to adjust the power at ~10 mW. Before behavioral testing, the ferrule implants on the top of the animals were coupled to splitter branching fiber-optic patch cords (Doric Lenses, Québec, Canada) with ceramic ferrule sleeves (Thorlabs, Inc, Newton, NJ, USA). Immediately after receiving a single train of laser pulses (50 ms/pulse × 10 pulses at 10 Hz), the mice were placed into the chambers for operant training.

## Optogenetic intracranial self-stimulation

Mice with intra-BLA injection of AAV-DIO-ChR2 or AAV-DIO-EYFP (control) were trained for oICSS in operant chambers (ENV-307W-CT, Med Associates Inc, St Albans, VT, USA), each equipped with two retractable levers on the wall. A fiber-optic rotary joint (FRJ_1x1_FC, Doric Lenses, Québec, Canada) mounted on the top of the exterior box was used to connect the splitter branching fiber-optic patch cords and the input patch cable attached to the laser system, allowing the cable-tethered mice to move freely during operant responding. To minimize the interference of laser flash with animals' behavior, ceramic sleeves connecting the ferrules and patch cord were covered with similarly sized black tubing. Scheduling of experimental events and data collection were accomplished using MED-PC IV software from Med-Associates Inc. The experiment consisted of the following sequential phases:

1. Initial conditioning: During a 1 hr session (fixed ratio 1 schedule), mice were allowed to move freely within the operant chambers. A single press on the active lever triggered the bilateral delivery of a train of laser pulses into the NAc, concurrent with the cue light illuminating above the active lever over the duration of laser stimulation (0.25 or 0.75 s). There was no timeout period during the training session. Pressing the inactive lever had no consequence. The criterion of ICSS was set to be a minimum of 100 active lever presses with the ratio of active/inactive lever presses ≥2:1 over the 1 hr session. The mice were first trained for three sessions with a weak

ICSS protocol (20 Hz, 30 times, pulse duration of 5 ms). During following sessions, the stimulation frequency was increased to 40 Hz with reduced pulse duration (1 ms; 10 mW; 10 times).

2. *Extinction*: After the cue-conditioned ICSS was established, mice were placed back into the operant chambers, where pressing the active lever only resulted in the presentation of cue without stimulation. Pressing the inactive lever did not result in stimulation or cue.

3. *Cue-induced ICSS seeking*: After retrained with the 40 Hz ICSS protocol for four sessions, the mice were kept within their home cages for 21 days. The mice were then placed back into the same operant chambers, in which pressing the active lever resulted in the presentation of cue without stimulation. Pressing the inactive lever did not have any consequence.

## Intravenous cocaine SA

The jugular vein catheterization was completed concurrently with the guide cannulae implantation 1 week before the cocaine SA training. Briefly, an incision was made in the neck to expose the right external jugular vein and an incision on the back for implanting a vascular access button (Instech Laboratories, Inc, Plymouth Meeting, PA, USA), which was connected to a silicone catheter (Dow Corning, 0.3 mm ID × 0.64 mm OD). Catheter placement was done by tunneling subcutaneously from the back incision and inserting the tubing into the jugular vein at the length of 1.1 cm. After the incisional areas and the button were secured by suturing, mice were singly housed in their home cages for recovery. The patency of the intravenous catheter was maintained by daily flush with heparinized saline (30 USP units/mL).

After ~7 days of recovery, catheterized mice were trained for cocaine SA in mouse modular operant chambers (ENV-307W-CT, Med Associates Inc, St Albans, VT, USA) under a fixed ratio 1 schedule. The chamber contained two retractable levers on one side of the wall and an LED cue light is located above the active lever. A 3 mL syringe containing 1.5 mg/mL cocaine solution was loaded to an infusion pump (PHM-200, Med Associate Inc) before each 2 hr SA session. Each session started by extending the two levers into the chamber and illuminating the house light (continuously on throughout each session). Intravenous injection was triggered by a single active lever press, enabling the mice to receive cocaine solution at a dose of 0.75 mg/kg/infusion and in a volume of 0.5 μL/g/infusion over ~2 s, depending on the bodyweight of the subject. The infusion was accompanied by a cue light illumination over the course of infusion. To prevent cocaine overdose, there was a 10 s timeout period between two consecutive infusions, during which pressing the active lever was recorded but did not result in cocaine infusion. Pressing the inactive lever had no programmed consequence.

For the test of cocaine seeking, mice underwent withdrawal after the 10-day cocaine SA procedure and were re-exposed to drug-associated cues in the same operant chambers on withdrawal days 7, 21, and 45. In the 1 hr cue-induced cocaine-seeking session, both the active and inactive levers were extended. Pressing the active lever resulted in the presentation of light cues without cocaine infusion, while pressing the inactive lever resulted in nothing. The number of active lever presses was used to assess cue-induced cocaine seeking.

## Chemogenetic manipulations of BLA-to-NAc pathways

The CNO solution was freshly prepared by dissolving the compound in saline and filled in a 10 μL Hamilton syringe (22 g blunt tip) connected to the customized internal cannulae (C315DCMN/SPC, P1 Technologies) by cannula-tubing (C313CT, P1 Technologies, Roanoke, VA, USA). To infuse CNO into the NAc, the mice were gently held in hand without anesthesia before the infusion. After the dummy cannula inserts were removed, needles were gently inserted into the cannulae. Saline or CNO (3 μM, 500 nL) was infused into NAc through the needle at 150 nL/min driven by a syringe pump (Pump 11 Elite, Harvard Apparatus). The injection needle was left in place for 5 min after injection to reduce backflow. The guide annulae were then recapped and mice were placed in operant chambers for testing.

## Tissue histology and imaging

Mice were anesthetized with a ketamine/xylazine mixture and then transcardially perfused with 0.9% saline, followed by 4% paraformaldehyde in 0.1 M phosphate buffer (BP, pH 7.4). After being removed from the skull and post-fixed in the same fixative overnight at 4°C, the brain tissues were embedded in 4% low melting point agarose (IBI Scientific, Dubuque, IA, USA) and sectioned coronally at room

temperature using vibratome (VT1200S, Leica, Wetzlar, Germany) at a thickness of 50 µm and collected in PB. Free-floating sections were mounted onto the slide for viewing the fluorescence signal. In some experiments, slices were processed with subsequent immunohistochemistry to enhance the signal of mCherry by antibodies: rabbit anti-RFP (1:500, 600-401-379, Rockland) and Alexa Fluor 568, Donkey Anti-Rabbit IgG (1:500, ab175470, Abcam). In this case, the floating tissues are incubated with the primary antibody in blocking buffer (4% bovine serum albumin+0.3% Triton X-100 in BP) at 4°C overnight and washed three times with PB, followed by 2 hr incubation with the secondary antibody at room temperature. Images were acquired using a Leica SP5 laser-scanning confocal microscope.

## Data acquisition and statistics

All results were expressed as mean ± s.e.m. Most experiments were replicated in 4–13 mice. All data collection was randomized. No statistical methods were used to predetermine sample sizes, but our sample sizes were similar to those reported in previous publications with similar experimental designs (*Xia et al., 2020*; *Wang et al., 2018*; *Ge et al., 2021*; *Wright et al., 2020*). All data were analyzed offline, and investigators were not blinded to experimental conditions during the analyses. Statistical analyses were performed in GraphPad Prism (v9) and SigmaPlot (13.0). Statistical significance was assessed using two-tailed two-way mixed or three-way ANOVA followed by posttests, as specified in the related text. Differences were considered significant when the p-value <0.05.

## Acknowledgements

We thank Jaryd Ross and Min Li for excellent technical support, and the Susumu Togegawa lab for providing the Rspo2-Cre and Cartpt-Cre mouse lines. The authors' work was partially supported by NIH grants, DA023206 (YD), DA040620 (YD), DA047861 (YD), DA051010 (YD). Cocaine chloride was provided by NIDA Drug Supply Program.

## Additional information

### Funding

| Funder | Grant reference number | Author |
|---|---|---|
| National Institute on Drug Abuse | DA23206 | Yan Dong |
| National Institute on Drug Abuse | DA40620 | Yan Dong |
| National Institute on Drug Abuse | DA47861 | Yan Dong |
| National Institute on Drug Abuse | DA51010 | Yan Dong |
| National Institute of Neurological Disorders and Stroke | NS107604 | Oliver M Schlüter |
| National Institute on Drug Abuse | DA43826 | Yanhua H Huang |
| National Institute on Drug Abuse | DA46346 | Yanhua H Huang |
| National Institute on Drug Abuse | DA46491 | Yanhua H Huang |
| National Institute on Alcohol Abuse and Alcoholism | AA28145 | Yanhua H Huang |

The funders had no role in study design, data collection and interpretation, or the decision to submit the work for publication.

## Author contributions
Yi He, Conceptualization, Data curation, Formal analysis, Investigation, Methodology, Writing – original draft, Writing – review and editing; Yanhua H Huang, Conceptualization, Resources, Supervision, Writing – review and editing; Oliver M Schlüter, Conceptualization, Resources, Funding acquisition; Yan Dong, Conceptualization, Resources, Supervision, Funding acquisition, Writing – original draft, Project administration, Writing – review and editing

## Author ORCIDs
Yi He http://orcid.org/0000-0002-9503-0565
Yan Dong https://orcid.org/0000-0003-0016-9028

## Ethics
This study was performed in accordance with the Guide for Care and Use of Lab animals of the National Institutes of Health. All animals were handled according to approved Institutional animal care and use committee protocol (22010508) of the University of Pittsburgh. The protocol was approved by the committee on the Ethics of Animal Experiments of the University of Pittsburgh (D16-00118). All surgery was performed under anesthesia and every effort was made to minimize suffering.

## Decision letter and Author response
Decision letter https://doi.org/10.7554/eLife.89766.sa1
Author response https://doi.org/10.7554/eLife.89766.sa2

# Additional files

## Supplementary files
• MDAR checklist

## Data availability
Raw data are presented as individual data points in all figures whenever available. Numerical raw data of all presented data are available as source data files, as well as at https://doi.org/10.5061/dryad.gqnk98stv.

The following dataset was generated:

| Author(s) | Year | Dataset title | Dataset URL | Database and Identifier |
|---|---|---|---|---|
| Dong Y | 2023 | Data from: ue- versus reward-encoding basolateral amygdala projections to nucleus accumbens | https://doi.org/10.5061/dryad.gqnk98stv | Dryad Digital Repository, 10.5061/dryad.gqnk98stv |

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
