## [Editor Report]

He et al.,2023 employ conditional genetics and in-vivo manipulations of neural activity to present valuable findings illustrating distinct functions of monosynaptic glutamatergic projections from R-spondin2-positive neurons and protein phosphatase 1 regulatory subunit 1B-positive neurons in the basolateral amygdala to the nucleus accumbens in cue-dependent operant conditioning for intracranial self-stimulation and cocaine rewards. While the evidence presented is solid and supported by well-designed experiments, the predictive capacity of the current model could be further enhanced by future experiments that delve into behavioral outcomes during operant conditioning when both types of projections are activated.

---

## [Decision Letter]

**Decision letter after peer review:**

Thank you for submitting your article "Cue- versus reward-encoding basolateral amygdala projections to nucleus accumbens" for consideration by *eLife*. Your article has been reviewed by 2 peer reviewers, one of whom is a member of our Board of Reviewing Editors, and the evaluation has been overseen by Michael Taffe as the Senior Editor. The following individual involved in review of your submission has agreed to reveal their identity: Burt Sharp (Reviewer #2).

*Reviewer #1 (Recommendations for the authors):*

Summary

Previous work had already identified the R-spondin2-positive(Rspo2) magnocellular neurons and the protein phosphatase 1 regulatory subunit 1B-positive (Ppp1r1b) parvocellular neurons, which project from the basolateral amygdala (BLA) to the Nucleus Accumbens ( NAc) BLA-NAc, in driving motivated behavior. The manuscript authored by He et al. introduces a nuanced perspective, distinguishing the roles of these two distinct cell populations in behavioral responses. Particularly, this research sheds light on their contributions to unconditioned stimulus (US)-mediated responses, which reinforce drug-taking behavior, and conditioned stimulus (CS), which promotes drug-seeking behavior. Specifically, the authors found that robustly optogenetically stimulating either the Rspo2 or Ppp1r1b presynaptic terminals in the NAc resulted in the promotion of US-mediated responses, such as intracranial self-stimulation (ICSS). However, only stimulation of Ppp1r1b neurons contributed to CS-mediated reinforcement responses, including cue-induced ICSS and cue-induced cocaine seeking following withdrawal from cocaine self-administration.

Strength

1. The manuscript is well-written and organized.

2. The authors establish a causal behavioral relationship by combining in-vivo physiology with genetic dissection.

Weaknesses

1. The differentiation between dopamine D1 receptor and dopamine D2 receptor-expressing NAc medium spiny neurons is determined by the presence or absence of D1-driven tdTomato. However, this approach has a limitation, as there exists a subset of neurons that express both D1 and D2 receptors, leading to their labeling with tdTomato as well.

2. Considering the relatively small sizes of the animal and recording samples in experiments 1L and 1X, it remains uncertain whether the authors possessed adequate statistical power to ascertain whether Rspo2 and Ppp1r1b neurons exhibited a discernible preference for innervation of the core or shell regions.

3. Data transparency is somewhat lacking in the manuscript, as it does often not explicitly state the number of animals used in each experiment, and individual data points are infrequently presented.

4. The manuscript exhibits deficiencies in adequately presenting the statistical analysis employed for the experiments. Within the text and/or figure legends, there is a lack of clarity regarding the specific statistical tests applied to accommodate repeated days or sessions, such as during ICSS and extinction. Furthermore, the effects in relation to the day of withdrawal are not explicitly specified. The incorporation of post-hoc analyses is sparse, and it remains unclear whether statistical tests for repeated measures were employed.

5. There appears to be an insufficient depth of discussion regarding the reasons behind the Rspo2 projections not affecting cocaine self-administration, despite their promotion of ICSS effects. This is particularly noteworthy considering the existing literature that highlights the behavioral contributions of Rspo2 neurons.

*Reviewer #2 (Recommendations for the authors):*

Neural projections from BLA to NAc have been implicated in both US and CS-dependent reinforcement in previous reports. The current experiments employ contemporary methods including opto- and chemogenetics to dissect the effects of two distinct neural projections from BLA to NAc on cue-dependent operant conditioning to ICSS and cocaine. The identification of contrasting functions for these two BLA glutamatergic projections to NAc median spiny neurons, both D1R and D2R, is a novel conceptual advance that may be relevant to the development of targeted therapeutics for addictive disorders. The Ppp1r1b+ BLA projection appears to be involved in the maintenance of longterm conditioning of the cocaine cue and in the incentive salience of this cue, whereas the Rspo2+ projection is involved in encoding the unconditioned reward. These studies would be strengthened by additional controls to further evaluate the function of the Ppp1r1b projection when both projections are concurrently activated during operant conditioning. Such a study might answer the following: Does a similar level of active presses resist extinction as observed following conditioning dependent only on activation of Ppp1r1b terminals? This could enhance the predictive value of the current model. The effects of additional extinction sessions, beyond the three studied in the ICSS model, might also be informative.

A. Please state the number of daily extinction sessions used during cocaine withdrawal in the d7, d21 and d45 reinstatement tests.

B. 500nl infusions into NAc are likely to infiltrate surrounding brain regions. Please show that the diffusion is limited to the NAc.

C. The statement in the Discussion regarding co-innervation of MSNs by both BLA projections needs further support. The results of random selection of MSN for electrophysiological evaluation in response to stimulation of each pathway could be due to alternative patterns of innervation: 1 pathway/1 MSN and/or 2 pathway/1MSN.

D. IF in fact the dominant innervation pattern is 2 pathways/1MSN, it is all the more important to understand the role of the Ppp1r1b pathway during extinction and reinstatement when both pathways are activated during operant conditioning.

E. Please discuss in greater detail the differences in the experimental models that yield opposite results regarding the role of BLA Rspo2+ neurons in appetitive vs aversive reinforcement. The current model largely excludes direct BLA-CEA projections by limiting activation to BLA Rspo2+ terminals in NAc. Hence, the current model may correctly identify the role of this pathway, but only in NAc-dependent appetitive reward.

F. The strong stimulation required of NAc terminals from both types of BLA projections for operant conditioning suggests, as the authors indicate, that other brain regions are concurrently activated by the laser stimulation of NAc. In this regard, VTA activation and release of DA in BLA may significantly affect PNs, especially the Ppp1r1b subset. Please discuss the implication of BLA DA-dependent modulation of the Ppp1r1b subset and the potential role of DA in incentive salience mediated by this BLA projection.

G. Although BLA Ppp1r1b projection fibers in NAc shell are more abundant than in core, as acknowledged by the authors, their data in Figure 1 (panels T and U) appears to suggest the converse: ChR2-eYFP fluorescence in Cartpt-core x D1-td Tomato mice is much greater in core fibers than shell.

---

## [Author Response]

Reviewer #1 (Recommendations for the authors):Strength1. The manuscript is well-written and organized.2. The authors establish a causal behavioral relationship by combining in-vivo physiology with genetic dissection.Weaknesses1. The differentiation between dopamine D1 receptor and dopamine D2 receptor-expressing NAc medium spiny neurons is determined by the presence or absence of D1-driven tdTomato. However, this approach has a limitation, as there exists a subset of neurons that express both D1 and D2 receptors, leading to their labeling with tdTomato as well.

We agree with the reviewer that our approach to identifying D1R versus D2R MSNs has limitation due to a subset of NAc neurons that co-express both D1R and D2R. However, the % of D1R-D2R co-expressing neurons is low, typically ~1.5%, 3.4%, or 6-7% depending on the measuring approach ^1^ (Bonnavion et al., 2022 bioRkiv). As such, we expect that the % of D1R-D2R-coexpressing neurons sampled as D1R neurons in the current study is very low and the related variability was averaged out.

2. Considering the relatively small sizes of the animal and recording samples in experiments 1L and 1X, it remains uncertain whether the authors possessed adequate statistical power to ascertain whether Rspo2 and Ppp1r1b neurons exhibited a discernible preference for innervation of the core or shell regions.

In both NAc core and shell neurons, we recorded reliable EPSCs evoked from both Rspo2 and Ppp1r1b projections. We agree with the reviewer that sample sizes were relatively small (n = 9, 6, 6 or 7), and because of this, we did not intend to make quantitative comparisons. We previously stated that “for each of the subprojections, we recorded MSNs from both the NAc shell and core and did not detect preferential innervation of core versus shell MSNs.”

We now change the statement to:

“For each of the subprojections, we recorded reliable EPSCs in MSNs from both the NAc shell and core, indicating that these subprojections form functional glutamatergic synapses on both NAc shell and core MSNs.”

3. Data transparency is somewhat lacking in the manuscript, as it does often not explicitly state the number of animals used in each experiment, and individual data points are infrequently presented.

We represented individual data points and reported both the number of recorded cells and number of animals (n/m, number of cells / number of animals) for all electrophysiological results. For behavioral results, because multiple sets of data were presented together over multiple days, there was no room to present individual data points within the graph. Alternatively, we provided the spreadsheets of individual data points in the Supplementary Material section for each figure panel such that readers can see individual data points if interested.

4. The manuscript exhibits deficiencies in adequately presenting the statistical analysis employed for the experiments. Within the text and/or figure legends, there is a lack of clarity regarding the specific statistical tests applied to accommodate repeated days or sessions, such as during ICSS and extinction. Furthermore, the effects in relation to the day of withdrawal are not explicitly specified. The incorporation of post-hoc analyses is sparse, and it remains unclear whether statistical tests for repeated measures were employed.

This concern might be due to a misunderstanding, because we provided rather detailed statistical reports for all figure panels. Here are two example reports typically appeared in figure legends:

Rspo2-to-NAc (active lever press, F_4,36_ = 1.2, p = 0.32; inactive lever press, F_4,36_ = 1.5, p = 0.22; Infusion, F_4,36_ = 1.5, p = 0.21; one-way ANOVA repeated measure; F). In this example, ANOVA test did not indicate a significant difference, so no posttest was performed.

Ppp1r1b-to-NAc (active lever press, F_4,20_ = 1.5, p = 0.25; inactive lever press, F_4,20_ = 0.4, p = 0.87; Infusion: F_4,20_ = 1.8, p = 0.16, Tukey posttest, G). In this example, ANOVA test indicated a significant difference, so posttest was performed.

5. There appears to be an insufficient depth of discussion regarding the reasons behind the Rspo2 projections not affecting cocaine self-administration, despite their promotion of ICSS effects. This is particularly noteworthy considering the existing literature that highlights the behavioral contributions of Rspo2 neurons.

We added a brief discussion in the revised version of the manuscript to reflect the following speculation about Rspo2 BLA neurons. Briefly, by manipulating somatic activities, previous studies demonstrate that Rspo2 BLA neurons contribute to motivated behaviors. In the current study, our chemogenetic manipulation was confined to the presynaptic terminals of the two subprojections within the NAc. As such, a lack of behavioral effects in response to Rspo2 presynaptic terminals suggests that the established behavioral contributions of Rspo2 BLA neurons are mediated by the projections of Rspo2 BLA to other brain regions.

Reviewer #2 (Recommendations for the authors):Neural projections from BLA to NAc have been implicated in both US and CS-dependent reinforcement in previous reports. The current experiments employ contemporary methods including opto- and chemogenetics to dissect the effects of two distinct neural projections from BLA to NAc on cue-dependent operant conditioning to ICSS and cocaine. The identification of contrasting functions for these two BLA glutamatergic projections to NAc median spiny neurons, both D1R and D2R, is a novel conceptual advance that may be relevant to the development of targeted therapeutics for addictive disorders. The Ppp1r1b+ BLA projection appears to be involved in the maintenance of longterm conditioning of the cocaine cue and in the incentive salience of this cue, whereas the Rspo2+ projection is involved in encoding the unconditioned reward. These studies would be strengthened by additional controls to further evaluate the function of the Ppp1r1b projection when both projections are concurrently activated during operant conditioning. Such a study might answer the following: Does a similar level of active presses resist extinction as observed following conditioning dependent only on activation of Ppp1r1b terminals? This could enhance the predictive value of the current model. The effects of additional extinction sessions, beyond the three studied in the ICSS model, might also be informative.

We thank the reviewer for the suggested experiments, which, as the reviewer indicated, will strengthen the conceptual model proposed in this study. We will incorporate these or similar experiments in our ongoing in vivo recording/imaging studies of the BLA and NAc in the context of cocaine seeking and extinction.

A. Please state the number of daily extinction sessions used during cocaine withdrawal in the d7, d21 and d45 reinstatement tests.

There was one session (1 h) of the extinction training on days 7, 21, and 45. This was stated in the Methods section.

B. 500nl infusions into NAc are likely to infiltrate surrounding brain regions. Please show that the diffusion is limited to the NAc.

We infused 500 nL of CNO solution into the NAc. It is technically difficult to trace this colorless chemical in the brain. Nonetheless, because CNO is a small molecule, it will diffuse to adjacent regions, with its concentration decreasing exponentially outside of the injection site. On the other hand, because CNO is expected to selectively affect DREADDs-expressing terminals enriched in the NAc. As such, it is a reasonable assumption that intra-NAc administration of 500 nL of CNO preferentially influenced DREADDs-expressing presynaptic terminals within the NAc.

C. The statement in the Discussion regarding co-innervation of MSNs by both BLA projections needs further support. The results of random selection of MSN for electrophysiological evaluation in response to stimulation of each pathway could be due to alternative patterns of innervation: 1 pathway/1 MSN and/or 2 pathway/1MSN.

We revised the statement to better reflect the results:

“In our experiments, stimulation of Rspo2- or Ppp1r1b-to-NAc presynaptic fibers evokes EPSCs in most randomly sampled MSNs, suggesting co-innervation of individual MSNs by both BLA subprojections. However, the low-throughput feature of electrophysiology did not allow to generate large sample sizes and, thus, did not rule out other possible innervation patterns.”

D. IF in fact the dominant innervation pattern is 2 pathways/1MSN, it is all the more important to understand the role of the Ppp1r1b pathway during extinction and reinstatement when both pathways are activated during operant conditioning.

We agree and we have hypotheses for this. Briefly, the Rspo2 and Ppp1r1b inputs generate different patterns of activation of NAc MSNs in behaving mice, and it is the different activity patterns that are correlate with different behavioral outputs. These experiments involve in vivo monitoring the neuronal activities in behaving mice and are being conducted in the lab for an independent project.

E. Please discuss in greater detail the differences in the experimental models that yield opposite results regarding the role of BLA Rspo2+ neurons in appetitive vs aversive reinforcement. The current model largely excludes direct BLA-CEA projections by limiting activation to BLA Rspo2+ terminals in NAc. Hence, the current model may correctly identify the role of this pathway, but only in NAc-dependent appetitive reward.

We revised the related discussion to improve clarity.

“By manipulating somatic activities or projections to the central nucleus of amygdala, previous studies demonstrate opposite roles of Rspo2 and Ppp1r1b BLA neurons in appetitive versus aversive reinforcement. Our current study focuses on the BLA-to-NAc projection and pinpoints the differential roles of two major BLA-to-NAc subprojections in invigorating US- versus CS-mediated reinforcement.”

F. The strong stimulation required of NAc terminals from both types of BLA projections for operant conditioning suggests, as the authors indicate, that other brain regions are concurrently activated by the laser stimulation of NAc. In this regard, VTA activation and release of DA in BLA may significantly affect PNs, especially the Ppp1r1b subset. Please discuss the implication of BLA DA-dependent modulation of the Ppp1r1b subset and the potential role of DA in incentive salience mediated by this BLA projection.

We thank the reviewer for raising this interesting topic. We have the expertise and interest to provide additional discussions along this line. However, after a few tries, we realize that it may not be the most appropriate opportunity for this discussion for two main reasons. First, our current study is highly focused on glutamatergic projections without data related to dopamine. Second, we indeed attempted to draft a coherent and balanced discussion about the activation of VTA dopamine neurons by BLA neurons, the VTA dopamine projection to BLA, and potentially differential regulation of Ppp1r1b versus Rspo2 neurons in the BLA. It was rather lengthy and read detached from the central theme of this study. We thus did not include the suggested discussion.

G. Although BLA Ppp1r1b projection fibers in NAc shell are more abundant than in core, as acknowledged by the authors, their data in Figure 1 (panels T and U) appears to suggest the converse: ChR2-eYFP fluorescence in Cartpt-core x D1-td Tomato mice is much greater in core fibers than shell.

We presented Figure 1T and U to show that the Rspo2 and Ppp1r1b neurons projected to both the NAc shell and core, echoing our electrophysiological observation that synaptic responses in both shell and core MSNs were detected in response to the stimulation or Rspo2 or Ppp1r1b subprojections. We did not intend to quantify the differential innervation intensity based on fluorescence because many variables were involved determining the fluorescence density in the current experimental setup. For example, both Rspo2 and Ppp1r1b neurons may follow the same anterior-posterior topographic projection rule from the BLA to NAc shell versus core (discussed in the manuscript), exhibiting biased innervation of the NAc shell versus core, regardless of the cell type.

1. Thibault, D., Loustalot, F., Fortin, G.M., Bourque, M.J., and Trudeau, L.E. (2013). Evaluation of D1 and D2 dopamine receptor segregation in the developing striatum using BAC transgenic mice. PLoS One 8, e67219. 10.1371/journal.pone.0067219.